**www.cambridge.org/qrd**

## Perspective

Darwinian evolution; protein folding; origin of life

**Corresponding author:**
Ken A. Dill;
Email: dill@laufercenter.org

# Origins of life: first came evolutionary dynamics

Charles Kocher[1,2] and Ken A. Dill[1,2,3]

[1]Laufer Center for Physical and Quantitative Biology, Stony Brook University, Stony Brook, NY, USA; [2]Department of Physics and Astronomy, Stony Brook University, Stony Brook, NY, USA and [3]Department of Chemistry, Stony Brook University, Stony Brook, NY, USA

### Abstract

When life arose from prebiotic molecules 3.5 billion years ago, what came first? Informational molecules (RNA, DNA), functional ones (proteins), or something else? We argue here for a different logic: rather than seeking a *molecule type*, we seek a *dynamical process.* Biology required an ability to evolve before it could choose and optimise materials. We hypothesise that the *evolution process* was rooted in the *peptide folding process.* Modelling shows how short random peptides can collapse in water and catalyse the elongation of others, powering both increased folding stability and emergent autocatalysis through a disorder-to-order process.

## Requirement for life's origin: persistent propagation

What was the origin of life? A pre-requisite for answering that question is to define the difference between dead and alive. Defining life has been notoriously challenging (Schrodinger, 1944; Cleland and Chyba, 2002; Popa, 2004; Benner, 2010; Machery, 2012; Pross, 2016; Plaxco and Gross, 2021). For example, the ability to metabolise, grow, and duplicate is not sufficient to distinguish life from a candle flame. Nevertheless, a good consensus definition is from NASA: 'Life is a self-sustaining chemical system *capable of Darwinian evolution*' (Joyce *et al.*, 1994). The italics are ours; the clear implication is that life could not begin before some dynamical adaptation process of molecules was already operative.

So, in order to seek the origins of life, we ask what physico-chemical process(es) could have driven prebiotic molecules to become autocatalytic and adaptive. We call this *The Day Two Problem*, to distinguish it from traditional questions about 'What Came First', which we call *The Day One Problem.*

- *The 'Day One' Question:* What material came first? Think of the metaphorical 'chicken or the egg problem' (although in the real world, the chicken-and-egg framing is misleading because many things, including chickens and eggs, emerged in parallel, not in series). For example: Did life start as an RNA World (Gilbert, 1986; Joyce and Szostak, 2018)? Or a lipid world (Segré *et al.,* 2001; Deamer, 2011), an amyloid world (Maury, 2009, 2015, 2018) or through metabolism first (Wächtershäuser, 1988; De Duve and De Neufville, 1991; De Duve, 1995; Cody, 2004; Shapiro, 2006; Jordan *et al.,* 2021; Matsuo and Kurihara, 2021)? Our view is that the full origin of life required many aspects – metabolism, information, protein-like catalysts and makers, and others – to arise together.
- *The 'Day Two' Question:* What dynamical process might have driven prebiotic molecules to become self-sustaining and self-serving? How might prebiotic molecules undergo directed change from Day One to Day Two and beyond? For biology to *operate*, it requires an *operating system.* What would drive molecules to evolve further on their own?

We give here a hypothesis. We describe a mechanism by which prebiotic undirected syntheses of short peptides could plausibly become 'makers', that is molecules that persistently make other molecules. We give a summary and perspective of recent computer modelling of a disorder-to-order process that achieves positive feedback against the forces of degradation. We start below by positing Darwinian evolutionary dynamics as a driven machine cycle of steps, because this vantage point illuminates principles about possible molecular origins.

## Description of Darwinian dynamics as a cyclic machine

Darwinian Dynamics has three well-known features: (1) Replication, moms making more moms; (2) Mutation, search and discovery of sequence → function polymers that create new molecular entities and mechanisms and (3) Selection, competition-driven upratcheting of fitness. Fig. 1 expresses these properties in the form of a machine-like, biosphere-wide nonequilibrium (NEQ) cycle that we call the *Darwinian Evolution Machine* (DEM). Described in detail elsewhere (Kocher and Dill, 2023), the cycle operates from left to right. (*a*) At time *t*, *X* indicates a wild-type (status quo) population. (*b*) A mutation occurs in some individual. (*c*) That individual cell is

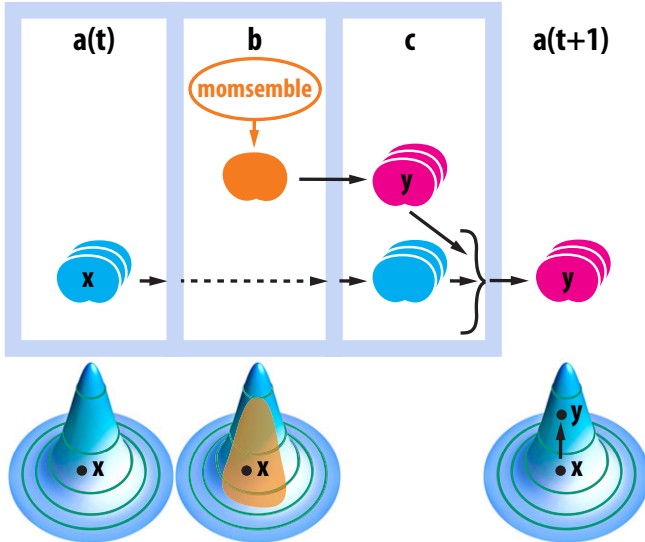

**Fig. 1.** Top: The DEM cycle. (*a*) At time *t* shows a population *X* of wild-type cells. (*b*) One cell mutates. That cell grows to have population *Y* in (*c*). Populations *X* and *Y* compete, one wins, and the cycle begins again as (*a*) at time *t*+1. Bottom: Fitness landscapes show the separation of actions. From point *X*, mutation entails a random, relatively unbiased exploration (orange region). The third landscape shows selection, in this case of *Y*, where the bias and preference occurs.

grown up into a population $Y$. Populations $X$ and $Y$ compete for resources. ($a(t+1)$) The winner becomes the new status quo wild-type population, thus becoming 'remembered' in the population, and gains more resources. The cycle repeats, driven by a persistent external supply of resources.

The steps (*b*) and (*c*) of population growth on the resources, competition and winning can be expressed as population-resource dynamics. For example, if we have $N$ competitors $A_n$, labelled by their phenotype, fighting for one resource $r$,

$$\frac{dr}{dt} = g(r) - D_r(r) - \sum_n U_n(A_n, r),$$

$$\frac{dA_n}{dt} = R_n\left(\frac{U_n(A_n, r)}{A_n}\right)A_n - D_n(A_n), \qquad (1)$$

where $g(r)$ describes the NEQ input of the resource, $D_r(r)$ and $D_n(A_n)$ are decay/death terms, $U_n(A_n, r)$ is the total resource use rate by moms of type $n$, and $R_n$ is the reproduction rate of one mom of type $n$ given that mom eats resource at a rate $U_n(A_n, r)/A_n$. Mutational discovery will introduce new moms, say $A_{N+1}$, which are then selected for or against by the DEM. The most competitive moms are 'remembered' by the DEM because they are good enough autocatalysts to maintain persistent populations. 'Fitness', which is the term that describes this competition-driven selection, is non-trivial to define, and can be model-dependent; see Supplementary Material SI.2 for more discussion.

Known biology constrains the mathematical form that is needed for the function $U$. Often, population genetics modelling approximates it as linear in both resources and moms, $U_n(A_n, r) = r k_n A_n$. But, such linearity leads to 'winner-take-all' (WTA) dynamics (Volterra, 1928; Fisher, 1930; Gause, 1934; MacArthur, 1970; Hsu *et al.*, 1977; Tilman, 1982; Chesson, 1990; Lifson, 1997; Pross, 2011; van Opheusden *et al.*, 2015), a dynamics that misses important features of evolution. Instead, evolution often gives 'peaceful coexistence' of multiple species on a given resource (Hutchinson, 1961; Armstrong and McGehee, 1980; Chesson, 2000; Chesson and

Kuang, 2008; Charlebois and Balázsi, 2016; Barabás *et al.*, 2018; Goyal *et al.*, 2018; Wang *et al.*, 2022). Peaceful coexistence is captured using a saturating function (Beddington, 1975; DeAngelis *et al.*, 1975; Novak and Stouffer, 2021; Stouffer and Novak, 2021; Kocher and Dill, 2023),

$$U_n(A_n, r) = \frac{r k_n A_n}{b_n + c_n r + A_n}, \qquad (2)$$

which simply expresses two natural limits, that maximum concentrations of moms are finite and that speeds of producing offspring are finite. What is novel in the present DEM perspective is the combining of this generalised form of population-genetics (Eq. (1)) with iterative mutation, competition and selection cycles (Kocher and Dill, 2023). In the following section, we extract four principles from this DEM perspective that helps us formulate possible molecular precursors in the next sections.

## Features of evolution that are relevant for origins

The DEM model perspective illuminates what is needed for the origin of life. First, the DEM is a *maker of makers*, a process of moms creating more moms. The DEM is an *autocatalytic set, or a collection of entities, each of which can be created catalytically by other entities, such that as a whole, the set is able to catalyse its own production* (Hordijk, 2019). An extensive literature describes the importance of autocatalytic sets in the origins of life (Eigen, 1971; Kauffman, 1971, 1986; Kauffman *et al.*, 1993; Dyson, 1999, 1982; Jain and Krishna, 2002; Hordijk and Steel, 2014; Hordijk, 2019; Hordijk *et al.*, 2022). The positive feedback of makers making makers contributes to the self-sustaining nature of evolution.

Second, environments that are unruly and fluctuating can sort winners from losers through booms and busts (Doebeli *et al.*, 2021; Wang *et al.*, 2022). Booms and busts drive the recycling of resources, taking resources away from the losers and giving them to the winners, thus driving the rich to get richer on the road to autocatalysis.

Third, DEM Dynamics can sustain *peaceful coexistence* among multiple agents at the same time. In a world of winners-taking-all (WTA), without peaceful coexistence, evolution would have been brittle, always on the edge of extinction. If $X$ is more fit than $Y$ in environment $E_1$, $X$ would be the lone survivor in a WTA model. Now, if the environment fluctuates to $E_2$, which kills $X$, then the whole ecosystem dies. Instead, in a world of coexistence, diversity preserves the ecosystem. An ecosystem that has an ensemble of *backup moms* is more robust to unpredictable new environments. Ensembles are crucial for long-term survival and persistence of the DEM.

And fourth, the biosphere-wide DEM is a *driven* machine: its cycles of molecule-making are powered by uptake of out-of-equilibrium resources from the environment. There are different tendencies for driven systems than for equilibrium processes. Some detail is given in Supplementary Material SI.1; here we just give a few examples. (1) A fluid subject to gravity will flow down a hill and stop in the valley at the bottom. But, a fluid subject to a strong force can flow beyond the valley to cross the next hill and beyond. (2) A TV set or computer performs intricate functions as long as it is 'plugged in'. Its current flows are not predicted by principles of equilibrium, such as the Second Law. Such devices only tend to equilibrium when they are unplugged. Think about an electromagnet. A metal rod will not pick up nails, but when a current is driven around the bar, it will. An electromagnet is driven by the current

input. Its action is a nonequilibrium (NEQ) force. That force goes to zero when the input current is turned off. (3) While equilibrium systems tend downhill in energy (or free energy), driven systems can also go uphill. Think of chemical reactions, like binding events or protein folding or molecular association or partitioning processes; under common circumstances, their stable states are predicted by tendencies towards minimum free energies. But, living systems have *biochemical cycles,* where uphill steps are driven by a coupling to downhill steps. The persistence of the DEM for 3.5 billion years is because biology has become so capable of exploiting the food, energy, and matter out-of-equilibrium aspects of its environments. How might the DEM have arisen from prebiotic molecular processes? Below are some of the key questions.

## Puzzles about the molecular origins of the DEM

- *What molecules were the first 'makers?'* Today's biological maker molecules are proteins and RNA, chain molecules that encode different functionalities as different monomer sequences. How did prebiotic polymers come to have sequence → function relationships? What simple molecular process started producing self-sustaining maker molecules?
- *How was molecule-making powered by external forces?* How did molecule–making come to outcompete molecule degradation and become so sustainable?
- *How did makers and catalysts become mobile, molecular-scale and editable?* An enormously transformative event in the prebiotic transition from chemistry to biology was the transition from catalysts that were immutable macroscale surfaces to microscale mobile editable proteins. Current thinking is that prebiotic reactions were first catalysed by mineral or clay surfaces (Wächtershäuser, 1988), or interfaces (Holden et al., 2022), or in hot volcanic vents (Martin et al., 2008). Such catalysts are macroscopic, geographically immovable, and fixed in their single-reaction catalysis under fixed conditions. But cells need whole biochemical pathways, where multiple reactions are strung together to achieve complex chemistry. Each step has a tailored catalyst that provides precisely the right acceleration of precisely that reaction, is mobile and small enough to fit inside a cell, and functions in the same water solvent as all the other requisite catalysts at room temperature. As a metaphor, consider the importance in the Industrial Revolution of steam engines, which replaced immovable energy sources of rivers and waterfalls by power that was mobile and tailorable to circumstances. How did prebiotic catalysis 'learn' to become untethered from rigid macroscopics to become flexible mobile microscopic biopolymers?
- *What was 'fitness' before there were cells?* Biological organisms are self-serving. This is captured in the multi-faceted notion of *fitness.* In contrast, molecules are not self-serving. How would molecules start becoming selected for or against?
- *Needles in haystacks and blind watchmakers: overcoming the infinitesimal probabilities.* Life's origin is often considered impossibly improbable, like finding a needle in a haystack, or finding a watch made by a blind watchmaker (Dawkins et al., 1996). But those arguments are based on models that assume many improbable steps happen independently. There is a problem with those models. Life's originating events were surely not independent: they were correlated. The key questions are: (1) What was the nature of those correlations? and (2) In what physical process does each step build on the advantages of the preceding steps to give cumulative long-term sustainability?

## The case for proteins and the folding process

On the one hand, our view is that even the earliest life requires multiple components – functional molecules like proteins, informational molecules like RNA/DNA, encapsulation like lipids, and on-board energy like the ATP; see Supplementary Material SI.3 and Carter and Kraut (1974) and Frenkel-Pinter et al. (2020). On the other hand, our goal here is more modest, namely just to explain the roots of evolution-like dynamics, how molecules became makers, and how maker molecules developed sequence-to-function relationships.

In principle, the first sequence-to-function maker molecules could have been either RNA or proteins. The pros and cons of the *RNA world hypothesis,* that RNA came first, are discussed elsewhere (Joyce et al., 1994; Atkins et al., 2011; Robertson and Joyce, 2012; Joyce and Szostak, 2018; Wills and Carter, 2018). Here, we postulate that proteins came first, both for reasons discussed in those references and because of the need to first establish some form of propagation dynamics. Here is a short summary. (1) Proteins are most of a cell's mass, so the differential growth rates of cell evolution are largely a matter of differential protein production. (2) Proteins are today's main maker molecules, catalysing the reactions of cell growth. (3) Proteins are unique in having sequence → structure → function relationships. Most other polymers, including most RNAs, do not. Proteins achieve their actions, functions and mechanisms by virtue of their *native molecular structures.* The folding code is primarily a hydrophobic (H) and polar (P) code, which other linear biomolecules do not have. Consequently, while some RNA molecules do fold uniquely and are catalysts, they are driven by different forces. Proteins are compact because they are dominated by hydrophobic tertiary interactions, whereas RNA molecules tend to be stringy because they are dominated by secondary-structure interactions of hydrogen bonding and base stacking. Moreover, because hydrogen bonds and base stacking are relatively sequence-independent, where chain slippage leads to many local minima in free energy, RNA folding landscapes are bumpier and less funnelled than protein landscapes (Chen and Dill, 2000). Thus, even RNAs that actually have folded structures tend to have multiple ones, and those structures are only weakly specified by RNA sequences. (4) Proteins' unique folded states make proteins good catalysts. Folded proteins are miniature solids. Being a solid is exactly what is needed to catalyse chemical reactions, because catalyst atoms need to hold their places long enough to assist the reaction. (5) A 20 amino acid alphabet spans a range of chemistries, so they catalyse a range of reactions. For these purposes, RNA molecules are not as good as proteins. Even where a given reaction can be catalysed by either proteins or RNAs, proteins are often better (Plaxco and Gross, 2021). (6) While some RNA molecules can self-copy, those molecules would need to have very low error rates in order to persist (Eigen, 1971; Jeancolas et al., 2020). The first copying machines would have to have had near-perfect fidelities. However, exact copying would be too brittle, for the same reasons we explained above that winner-takes-all (WTA) competitions are: without a way of generating diversity, exact copying is too prone to extinction in the face of environmental changes. In our view, prebiotic forces did not aim at self-copying; they aimed instead towards becoming autocatalytic *sets,* not strict autocatalysts. Variance is crucial. Progeny must not be identical to moms. The origins

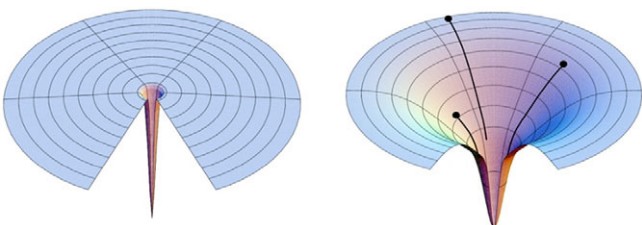

**Fig. 2.** Different landscapes of stochastic exploration: Golf courses *versus* Funnel Landscapes. Lateral directions are sampling degrees of freedom; the up-down direction is some measure of value (more value is downhill). (Left) *Blind Watchmaker, Needle-in-a-Haystack*: all states, except one, have no value. Success is nearly impossible. (Right) *Funnel Landscape:* From any starting point, there is often some direction that gives incremental advantage. And there are many routes for chaining together small advantages to more global advantage (black lines). Success is nearly inevitable. We believe evolution, once it gets going, is more funnel-like.

process must have some aspects of replication that are also to some degree unfaithful.

In terms of a dynamical process, protein folding has pertinent features. Protein folding entails a probabilistic needle-in-a-haystack search challenge through a disorder-to-order transformation. The folding search problem is now well understood in terms of *funnel-shaped energy landscapes* (Chan and Dill, 1991; Dill and Shortle, 1991; Wolynes *et al.,* 1995; Onuchic *et al.,* 1997; Wolynes, 1997; Dill *et al.,* 2007, 2008; Thirumalai *et al.,* 2010; Rollins and Dill, 2014; Nassar *et al.,* 2021). Fig. 2 compares a funnel landscape to a 'golf-course' landscape, which is premised on the assumption of uncorrelated independent events. 'Funnel' refers to the coarsest level of kinetic features, and not the potentially many finer-grained kinetic traps. Protein folding occurs so rapidly and towards such a unique ordered state because small random local steps combine together to lead effectively to the native state. In short, many proteins fold by rapidly finding needles in haystacks and creating complex watchmaker-like structures through small random correlated actions following combinatorially many microscopic routes via opportunistic chemical preferences. Protein folding gives both a metaphor for needle-in-a-haystack searching and a specific physical process, as described below, that could have become evolutionary dynamics.

## Emergent autocatalysis from HP foldcats

Here is our hypothesis, first in overview, then in more detail. We postulate that prebiotic syntheses could produce short peptide

chains, some of which collapse into compact structures in water because of their hydrophobic content. A fraction of those collapsed chains will have exposed hydrophobic surfaces, active as a primitive catalytic site, slightly accelerating the binding and elongation of other peptides. Computer simulations show that this mechanism leads to autocatalytic sets. The premise that amino acids could be produced and could polymerise into short random peptide chains under plausible prebiotic conditions is well-established (Miller and Urey, 1959; Wächtershäuser, 1988; Botta and Bada, 2002; Johnson *et al.,* 2008; Lambert, 2008; Ikehara, 2014; Foden *et al.,* 2020; Frenkel-Pinter *et al.,* 2020; Muchowska *et al.,* 2020; Holden *et al.,* 2022; Krasnokutski *et al.,* 2022).

However, existing peptide synthesis experiments do not explain how chains could have become long enough to fold and function like proteins; how they could become catalysts and makers; how the process could become autocatalytic; how they could give non-random sequence → structure relationships; how catalysis became mobile; or what are the molecular origins of fitness. We address these below.

Fig. 3 illustrates the chain elongation challenge. Typical polymer syntheses give mostly only short chains that are not long enough to fold and function as today's proteins do. However, it has been found in computer modelling that some heteropolymers behave differently (Guseva *et al.,* 2017). Chains that have particular sequences of hydrophobic (H) and polar (P) types of monomers, called HP polymers, collapse in water into compact states due to the hydrophobic effect. Even some relatively short sequences can collapse. Here, we call those chains *foldamers*. Furthermore, a small fraction of foldamer sequences can act as primitive catalysts, described below.

## HP chains can fold, catalyse and elongate

Fig. 4 shows that HP chain molecules have three general classes of behaviour in water, depending on their sequence of H and P monomers. (1) Some chains do not fold at all (think of the all-P sequence, for example). (2) Some HP sequences are foldamers, compact with hydrophobic cores. And (3) a fraction of HP foldamers happen to have surface patches that are concentrated in hydrophobic monomers; we call these surface regions 'landing pads', because these are regions that are sticky for other hydrophobic molecules floating in solution, Landing pads can be regions of catalysis. We call collapsed chains having landing pads *foldcats*, short for foldamer-catalysts.

These landing pads on foldcats could catalyse the covalent elongation of other 'client' HP sequences. The mechanism of this

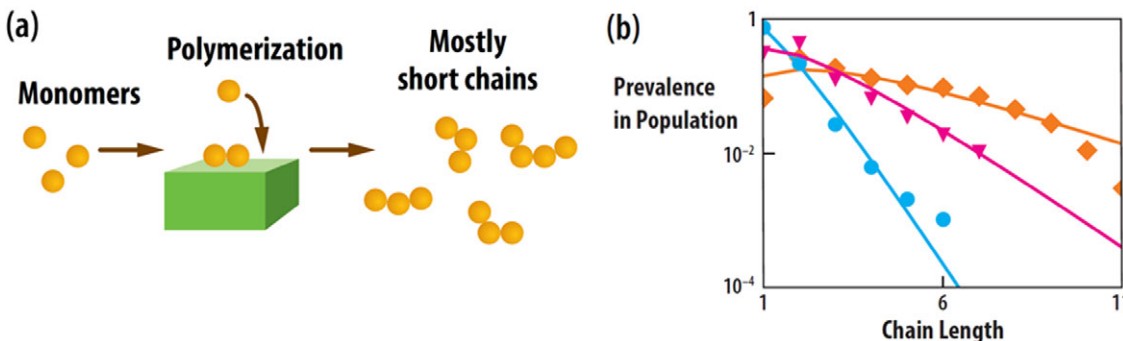

**Fig. 3.** Traditional polymerizations give mostly short chains, described by the Flory distribution. (*a*) A stationary catalyst polymerisation scheme. (*b*) Examples of the resulting Flory distribution, fit to experimental data: Orange (Kanavarioti *et al.,* 2001), pink (Ferris, 1999), and blue (Ding *et al.,* 1996). Populations rapidly diminish exponentially with chain length. This plot was reprinted with permission from Guseva *et al.* (2017).

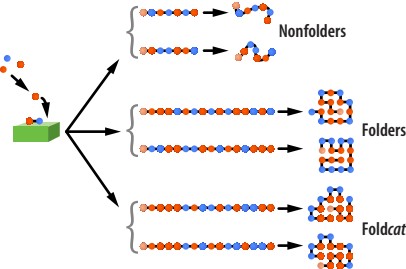

**Fig. 4.** Some HP chains can fold in water. HP chains are heteropolymers of hydrophobic (H, red) and polar (P, blue) monomers. Some sequences will not fold (nonfolders), while others will fold into compact states in water (*foldamers*), and a fraction of those sequences will have surface sites that can catalyse other reactions (*foldcats*).

catalysis process is shown in Fig. 5. Each foldcat sequence balls up, leaving a sticky spot (clustered H monomers) on its surface. A different peptide chain, call it a *client*, lands with its H monomers binding hydrophobically to the landing pad of the foldcat. A free H monomer from solution also lands on the landing pad. The spatial colocalization of the H monomer adjacent to the client chain can reduce the kinetic barrier to elongation of the client chain. The foldcat's job is to keep all the required pieces for elongation (the growing chain and a free monomer) in the same place. Peptide bond formation has a transition state barrier of 18 kcal mol$^{-1}$ (Gindulyte *et al.*, 2006). Spatial localization of two reactants, often called *proximity effects* or *enhanced effective concentrations*, is known to accelerate covalent bond formation reactions by as much as $10^8$ (Menger and Nome, 2019). For illustration, we have supposed only a binary code and only hydrophobicity-based landing pads. More realistically, a code will have more than two amino acid types, and more diverse interactions. The expectation that prebiotic peptides would have had both H and P amino acids is supported by the Miller–Urey experiment and recent variants of it (Miller and Urey, 1959; Botta and Bada, 2002; Johnson *et al.*, 2008). A proposed

minimal set would be GADV peptides (Ikehara, 2014), although there would be value in including cysteine (Foden *et al.*, 2020), and lysine or arginine for breadth of chemistry and control of aggregation.

This HP foldcat mechanism has recently been observed and explored in computer simulations (Guseva *et al.*, 2017). First, note that all these effects would likely have been almost negligibly small at first. Foldamers constitute only a fraction of all HP sequences; foldcats constitute an even smaller fraction; and colocalization-based rate enhancements are unlikely to be greater than a few kT in free energy (based on hydrophobicity estimates). But, it is not the *smallness* of populations or actions that matter. Rather, it is whether one step to the next entails some form of *systematic positive cooperativity*. What matters for origins (as well as for evolution in general) is whether some sub-population, even a very small one, is capable of some action – call it *emergent behaviour* – of positive feedback, so that it grows relative to other sub-populations, ultimately overcoming the relatively fixed forces of degradation. In general, a big challenge in origins-of-life research is that the initial seeding event is likely to be a very small signal in very large noise – precisely the sort of event for which devising a good experiment is difficult. Below, we describe how the HP foldcat mechanism predicts such emergent behaviours.

### How the folding process leads to the evolution process

Here are the emergent behaviours of the HP folding and catalysis mechanism.

- *Emergence of makers, catalysts and molecular functionalities.* From the short random peptides that are plausibly synthesised prebiotically, the HP foldcat mechanism produces longer chains; see Fig. 6a. On average, longer HP chains are more stably folded and more protein-like (because they bury more hydrophobic surface). So, as long as amino acids are input, the HP foldamer mechanism pushes from peptides towards

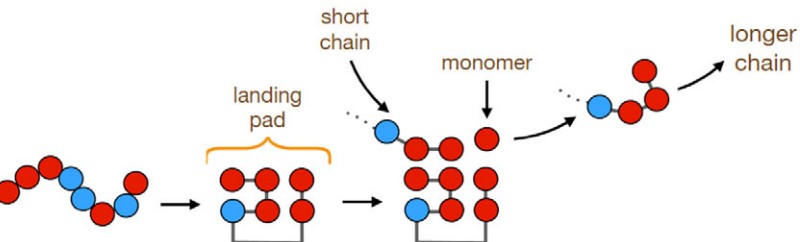

**Fig. 5.** HP foldcat chains can catalyse the elongation of other peptides. From the left: an HP sequence folds and exposes a hydrophobic surface ('landing pad'), a site on which a different chain can land and add a monomer to grow longer.

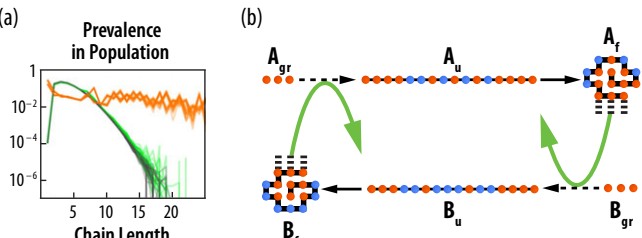

**Fig. 6.** The HP Foldcat mechanism: (a) grows longer chains and (b) populates an autocatalytic set. (a) Starting from random short HP molecules, chains elongate (orange) more than in the traditional Flory distribution (black) or the case of foldamers only with no catalysis (green). (b) Active, folded foldcat sequences ($A_f$ and $B_f$) amplify the populations of other growing foldcats ($A_{gr}$ and $B_{gr}$) while they are unfolded ($A_u$ and $B_u$), leading to an *autocatalytic set*.

proteins, creating more catalytic power and functional diversity. These foldcats are makers that make makers. An alternative mechanism proposed for chain elongation is templated ligation, but it requires enzyme assistance (Tkachenko and Maslov, 2015; Kudella *et al.*, 2021).

- *Emergence of sequence-to-function relationships.* This mechanism amplifies the populations of foldamers and foldcats, simply because foldcats are a larger proportion of longer-chain sequences. Foldcats form an autocatalytic set; Fig. 6b. Such situations, where some sequences are populated selectively relative to other sequences based on their functionalities, are the basis for sequence-to-function relationships.

- *Emergence of programmable mobile molecular machines.* Presumably, the first prebiotic peptides were synthesised on macroscale catalysts, fixed in space and inflexible in their actions. But, the foldcat mechanism then produces its own catalysts, poor at first and better later. This untethers the peptide catalysis process from fixed spaces. Now, catalysts are at the microscale: they are mobile; and they are diverse and programmable by virtue of the sampling of sequence space. We regard this untethering, from macro to micro, from fixed to mobile, to have been a transformative step from prebiotic chemistry to biology.

- *Emergence of adaptation.* Arguably, evolution's central principle is that organisms adapt to environments. Evolution's great power of innovation and resourcefulness comes from its mutational search, competition, and fitness-based selection. The HP foldamer perspective posits that such adaptivity could have originated from a disorder-to-order process, in which chain molecules sample different sequences; molecules compete for limited resources; and winners are those that are more stable and get more resources.

- *What is 'fitness' among molecules? First, just persistence.* Darwinian evolution chooses winners and losers based on fitness ratcheting. What are winners and losers in a prebiotic world of molecules? HP chains persist in stably folded states for longer or shorter times, based on their sequences. Longer chains are more stable because they bury more hydrophobic residues upon folding, and because compactness limits access to chemical agents that hydrolyze proteins. In unruly environments of booms and busts, molecules that are more stable persist by scavenging the recycled monomers and peptides from molecules that are less stable.

- *Emergence of a tipping point from error catastrophes to success catastrophe.* Prebiotic molecules are subject to degradation. Error catastrophes are unavoidable in direct-replicator mechanisms (Eigen, 1971; Jeancolas *et al.*, 2020). Short peptides will hydrolyze to monomers. The origin of life was a tipping point from error catastrophes (where degradation dominates), to a 'success catastrophe', where maker molecules establish persistent populations. Beyond this point, evolution and growth then prevail over degradation. Three factors explain this tipping point in the HP foldamer model: *(1) Autocatalysis.* As noted above, peptides grow longer, more stably folded, and form an autocatalytic set. This contributes positive cooperativity towards self-sustainability. *(2) A driven machine.* Like a TV set that is 'plugged in', the HP foldamer mechanism is driven by a persistent input. The input is amino acids (and at early stages, also a catalyst of peptide synthesis). It does not matter that most product peptides fail and degrade; being 'plugged in' means that the system keeps pumping to push the chain lengths higher.

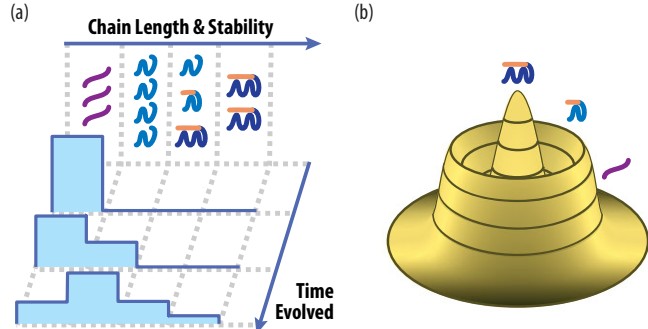

**Fig. 7.** The HP foldcat mechanism spontaneously grows populations of longer chains. (*a*) Longer proteins fold more stably and are more persistent to fluctuating environments. (*b*) A conceptual fitness landscape based on time of persistence of folding stability.

*(3) Adaptivity.* Autocatalysis and input power alone are not sufficient. Environments are unruly. Biology would not have survived without adaptability to changing environments. The combination of these factors contribute to a drive to ratchet up persistence over time; see Fig. 7.

To summarise, the HP foldamer mechanism explains how peptide synthesis and folding could result in the emergence of evolution-like propagation; see Table 1. But to be clear, we regard this not as the origin of life itself, but rather only as a precursor to it. Origins surely required much more than this: cell-like encapsulation, information and heritability, and more (some further discussion is given in the Supplementary Material).

## Evidence supporting the HP foldcat mechanism

Although there is no direct experiment testing this foldcat mechanism, several of its components are supported by experiments. *HP chains can fold and function.* The binary HP code dominates protein folding (Lim and Sauer, 1989; Bowie *et al.*, 1990; Kamtekar *et al.*, 1993; Dill *et al.*, 1995; Dill and MacCallum, 2012; Koga *et al.*, 2020). But also, a biomolecule backbone is not required; HP peptoids can fold and function too (Lee *et al.*, 2005; Yoo and Kirshenbaum, 2008). *Some random peptide sequences can fold.* It is not an infinitely dilute space of sequences that can fold. As discussed in Guseva *et al.* (2017), for HP chains up to length 25, 2.3% fold to unique structures and 12.7% of those foldamers, or 0.3% of all sequences, have the foldcat catalytic surface. *Peptide syntheses occur naturally.* Even in interstellar space, 6–8-mer peptides have been found (Kebukawa *et al.*, 2022; Krasnokutski *et al.*, 2022). Sea spray

**Table 1.** Correspondence between properties of evolution and properties of origins, in the HP foldamer model described in the text

| DEM | Origins |
| --- | --- |
| Moms | Protein molecules |
| … make moms | … make foldcats |
| Mutational search | Random sequences |
| Fitness | Persistence |
| Fitness gain | Folding stability |
| Degrees of freedom | Chain length |

or air-water surfaces could catalyse small peptide formation (Griffith and Vaida, 2012; Deal *et al.*, 2021; Holden *et al.*, 2022). *Some short peptides can catalyse reactions* (Adamala and Szostak, 2013; Rufo *et al.*, 2014). *Hydrophobic patches are common on proteins* (Lijnzaad *et al.*, 1996; Tonddast-Navaei and Skolnick, 2015), which we call landing pads in the foldamer mechanism. *Some proteins are synthesised without ribosomes* (Finking and Marahiel, 2004; Miller and Gulick, 2016).

## Outlook: from protein folding to evolution

We have posited that the origins of life could not have arisen without first a Darwin-like propagation mechanism. We believe function came before information, because we know of no driving force for the reverse. Rather than *genes using proteins to make new genes* (as in the Selfish Gene hypothesis Dawkins, 1976), our view is that *proteins use genes to make new proteins*. And, the foldcat mechanism indicates a way that the middleman – the gene – was simply not needed at first. This mechanism is based on solution physics – the oil–water and hydrogen-bonding forces of protein folding, the ability of miniature solids having different chemical moieties to catalyse reactions, and the ability of random syntheses to find and retain useful sequences based on their persistences. The foldcat mechanism addresses an important problem of origins research: it does not require a guiding hand of a researcher who chooses molecules, systems or processes. Instead, the foldcat mechanism is a disorder-to-order transition that bootstraps functional advantages that it finds from random search.

**Open peer review.** To view the open peer review materials for this article, please visit http://doi.org/10.1017/qrd.2023.2.

**Supplementary materials.** To view supplementary material for this article, please visit http://doi.org/10.1017/qrd.2023.2.

**Acknowledgements.** We thank Luca Agozzino and Gabor Balazsi for early discussions and Charlie Carter for extensive insightful comments. We are grateful to the Templeton Foundation and the Laufer Center for their support.

**Author contribution.** All authors contributed equally to this work.

**Financial support.** This work was supported by the John Templeton Foundation (grant ID 62564).

**Competing interest.** The authors declare none.

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
