## [Reviewer Report]

*Comments to Author*: The Authors Present Their Work “Origins Of Life: First Came Evolutionary Dynamics” where they present a Darwinian Evolution Machine (DEM) that explains the evolution of a population where mutations are possible, which in turn create differential populations that will compete for resources and will lead the system to have one winner population that takes the system to a new equilibrium where that population wins and becomes dominant. However, it is important to note that their DEM takes into account the possibility of peaceful coexistence between different populations instead of a winner-takes-all model. Peaceful coexistence would provide evolution with the robustness needed to be possible, otherwise extinction would have been most likely the rule. In addition, the DEM is not a closed system. Instead the DEM self-sustains due to the uptake of external resources.

Given the previous definitions the authors navigate in the description of which types of molecules would have had the necessary properties to be the protagonists in such a DEM model.

I think that the introduction is very well structured as well as the initial explanations that lead the author to the understanding of the properties the DEM, first makers need to have.

I think that the logical steps to explain how certain molecules become makers and these in turn develop sequence to function relationships is really appealing. It is also very interesting that the hypothesis that RNA and its ability to replicate is not the important property for this beginning of makers’ emergence but the ability of dynamic propagation by autocatalytic molecules. Furthermore the way in which the authors link the previous introduction of terms and concepts with the funneled energy landscapes theory is very nice.

To be honest, I have liked the paper a lot and since it is more of an hypothesis that builds up on several of the authors’ previous publications I do not have much to correct.

I do have some questions though.

The authors say that the evolutionary landscape that led to the initial pre-protein molecules was somehow funneled and not a golf-course. What I understand from this analogy is that if the landscape, because of the biophysical properties of the early makers, was already funneled and therefore success was nearly inevitable because of that (the authors say in Fig2 legend that they believe evolution is more funnel-like). If this is true, do you think then that the probability of having protein-like systems in planets like planet earth are almost certain, given that that would already set up the conditions to funnel evolutionary landscapes? What the consequences of this theoretical framework would be for the likely development of life or at least of early makers outside of earth?

The authors provide an explanation of the emergence of foldamers and foldcats based on a hydrophobic and polar set of amino acids. Can the authors maybe hypothesize or reflect on what kind of alphabet would be needed and compatible with the emergence of foldamers and foldcats. Do they have any preference towards some study that has hypothesized about the size of the primitive alphabet for when proteins emerged?

Minor comments

There is a mention to Fig3 which has panels a and b. But there is no mention of the individual panels. Panel b is difficult to understand. I found that a similar figure is reported in Guzeva et al, PNAS 2017. There the figure is better explained (also there is another panel) where the colors are meaningful. I would improve that figure panel to make it easier to understand. I would also place the y-axis next to the y-axis instead of in the corner. I was confused for a bit about if it was a title or an axis label.

In fig 4. There is no mention of the non folders in the legend as there are for the others.

Both Figs4 and 5 would benefit from saying which color is the H and which one is the P type of amino acid. I know it is a conceptual figure, but still I tried to mentally make sense of it and it became hard for a while.

Regarding Fig5 and the explanation from lines 186 to lines 193.. I did not fully understand it. I kind of understand the mechanism but somehow some details are escaping from my understanding. In particular I don’t understand how “These landing pads on foldcats could catalyze the COVALENT elongation of other client HP sequences. Maybe a better description in the conceptual schema form Fig5 would help or a modification of Fig5 to make it clearer.

In Fig6. There is no explanation to the labels in panel b (Agr, Au, Af…, etc). What are they?

This is just a suggestion that in my opinion would close the paper in a nicer way. I am missing a connection between the title and the latest part of the OUTLOOK section. The title states that “First Came evolutionary dynamics”. Although I understand what the authors refer to along the manuscript.. I final punch going back to that concept in a more explicit way would be really nice to close the paper.

There is a repeated “the” at line 117 in page 4.

Fig8 appears in the supplementary. Maybe it should be named FigS1?

The rest of the paper after Fig5 and its reference goes very smoothly and is nicely structured making its interpretation really straight forward.

---

## [Reviewer Report]

*Comments to Editor*: Hi Giulia, I am happy to review this manuscript (sorry that I clicked the “decline” button by mistake). Best wishes,-- Shi-Jie

*Comments to Author*: This manuscript presents a novel study of evolutionary dynamics. The authors first provided an in-depth description of the Darwinian Evolution Machine (DEM) cycle and its advantages for modeling the origins of life. The authors then discussed the puzzles about the molecular origin of DEM and concluded that proteins have essential features for being the maker molecules. Last, the authors presented the HP foldamer mechanism and demonstrated that short HP peptide chains can fold and catalyze the elongation of other peptides, resulting in an autocatalytic set and an evolution-like propagation.

This is an excellent manuscript with robust findings and interesting conclusions. I have only a few minor comments:

1. The figure captions lack sufficient details and do not fully describe the information presented in the panels. More detailed descriptions are necessary. Below are a few specific questions:

Fig. 3: In panel (b), what are the different lines and points representing?

Fig. 4 and Fig. 5: Why are the red and blue dots there? H monomers or P monomers?

Fig. 6: In panel (b), what do the symbols (Au, Af, etc.) mean??

2. The relationships between the different sections are not clear. Table I may be moved to the main text and further discussion of the correspondence between DEM cycle and HP foldamer model may be provided.

3. Page 4, Line 141. “Folded proteins are nanoscale solids, different for different sequences. In contrast, because RNA molecules are stiff and hydrogen bonded, they tend to be stringier - less folded, less compact and more floppy - and with poorly defined sequence-to-structure relationships.”

Structures are generally more conserved than the sequences and different RNA sequences can fold to the same or similar structures. Indeed, RNAs are usually more flexible. It would be useful for the authors to clarify the meaning of “poorly defined sequence-to-structure relationships” for RNAs.

4. Page 4, Line 144. “Accurate self-copying is not the property we seek here.”

Why is an accurate replication of informational polymers not a key property for the autocatalytic system? More discussions would be helpful here.

5. Page 4, the last paragraph describes the “golf course” landscape and an idealized “funnel” landscape for fast protein folding. What about more complex funnel landscapes, such as a global funnel-like landscape that involves kinetic traps and bumps? Can complex funnel landscapes also result in disorder-to-order transformation?

6. Page 5, Line 173. “However, it has been found in computer modeling that some heteropolymers behave differently.”

Citations of references are needed here.

7. Page 6, Line 186. “These landing pads on foldcats could catalyze the covalent elongation of other “client” HP sequences.”

In the HP foldamer mechanism, how long can a sequence be elongated? How does the covalent elongation occur when the H monomer is close to the landing pads on the foldcats?

8. The key idea behind the HP foldamer mechanism comes from the hydrophobic effects, where the hydrophobic “landing pads” can attract peptides and catalyze polymerization.The “landing pads” idea is quite interesting and may be generalized in the discussion. For example, nucleotides in a loop region of a folded RNA may also serve as the “landing anchors” for other nucleotides/short chains through base pairing.

---

## [Reviewer Report]

*Comments to Editor*: PLEASE LEAVE THIS FIELD BLANK AND USE THE ‘INSTRUCTIONS TO EDITORIAL OFFICE’ TEXT BOX

Reviewer, Shi-Jie Chen: Hi Giulia, I am happy to review this manuscript (sorry that I clicked the “decline” button by mistake). Best wishes,-- Shi-Jie

*Comments to Author*: Reviewer #2: The Authors Present Their Work “Origins Of Life: First Came Evolutionary Dynamics” where they present a Darwinian Evolution Machine (DEM) that explains the evolution of a population where mutations are possible, which in turn create differential populations that will compete for resources and will lead the system to have one winner population that takes the system to a new equilibrium where that population wins and becomes dominant. However, it is important to note that their DEM takes into account the possibility of peaceful coexistence between different populations instead of a winner-takes-all model. Peaceful coexistence would provide evolution with the robustness needed to be possible, otherwise extinction would have been most likely the rule. In addition, the DEM is not a closed system. Instead the DEM self-sustains due to the uptake of external resources.

Given the previous definitions the authors navigate in the description of which types of molecules would have had the necessary properties to be the protagonists in such a DEM model.

I think that the introduction is very well structured as well as the initial explanations that lead the author to the understanding of the properties the DEM, first makers need to have.

I think that the logical steps to explain how certain molecules become makers and these in turn develop sequence to function relationships is really appealing. It is also very interesting that the hypothesis that RNA and its ability to replicate is not the important property for this beginning of makers’ emergence but the ability of dynamic propagation by autocatalytic molecules. Furthermore the way in which the authors link the previous introduction of terms and concepts with the funneled energy landscapes theory is very nice.

To be honest, I have liked the paper a lot and since it is more of an hypothesis that builds up on several of the authors’ previous publications I do not have much to correct.

I do have some questions though.

The authors say that the evolutionary landscape that led to the initial pre-protein molecules was somehow funneled and not a golf-course. What I understand from this analogy is that if the landscape, because of the biophysical properties of the early makers, was already funneled and therefore success was nearly inevitable because of that (the authors say in Fig2 legend that they believe evolution is more funnel-like). If this is true, do you think then that the probability of having protein-like systems in planets like planet earth are almost certain, given that that would already set up the conditions to funnel evolutionary landscapes? What the consequences of this theoretical framework would be for the likely development of life or at least of early makers outside of earth?

The authors provide an explanation of the emergence of foldamers and foldcats based on a hydrophobic and polar set of amino acids. Can the authors maybe hypothesize or reflect on what kind of alphabet would be needed and compatible with the emergence of foldamers and foldcats. Do they have any preference towards some study that has hypothesized about the size of the primitive alphabet for when proteins emerged?

Minor comments

There is a mention to Fig3 which has panels a and b. But there is no mention of the individual panels. Panel b is difficult to understand. I found that a similar figure is reported in Guzeva et al, PNAS 2017. There the figure is better explained (also there is another panel) where the colors are meaningful. I would improve that figure panel to make it easier to understand. I would also place the y-axis next to the y-axis instead of in the corner. I was confused for a bit about if it was a title or an axis label.

In fig 4. There is no mention of the non folders in the legend as there are for the others.

Both Figs4 and 5 would benefit from saying which color is the H and which one is the P type of amino acid. I know it is a conceptual figure, but still I tried to mentally make sense of it and it became hard for a while.

Regarding Fig5 and the explanation from lines 186 to lines 193.. I did not fully understand it. I kind of understand the mechanism but somehow some details are escaping from my understanding. In particular I don’t understand how “These landing pads on foldcats could catalyze the COVALENT elongation of other client HP sequences. Maybe a better description in the conceptual schema form Fig5 would help or a modification of Fig5 to make it clearer.

In Fig6. There is no explanation to the labels in panel b (Agr, Au, Af…, etc). What are they?

This is just a suggestion that in my opinion would close the paper in a nicer way. I am missing a connection between the title and the latest part of the OUTLOOK section. The title states that “First Came evolutionary dynamics”. Although I understand what the authors refer to along the manuscript.. I final punch going back to that concept in a more explicit way would be really nice to close the paper.

There is a repeated “the” at line 117 in page 4.

Fig8 appears in the supplementary. Maybe it should be named FigS1?

The rest of the paper after Fig5 and its reference goes very smoothly and is nicely structured making its interpretation really straight forward.

Reviewer #3: This manuscript presents a novel study of evolutionary dynamics. The authors first provided an in-depth description of the Darwinian Evolution Machine (DEM) cycle and its advantages for modeling the origins of life. The authors then discussed the puzzles about the molecular origin of DEM and concluded that proteins have essential features for being the maker molecules. Last, the authors presented the HP foldamer mechanism and demonstrated that short HP peptide chains can fold and catalyze the elongation of other peptides, resulting in an autocatalytic set and an evolution-like propagation.

This is an excellent manuscript with robust findings and interesting conclusions. I have only a few minor comments:

1. The figure captions lack sufficient details and do not fully describe the information presented in the panels. More detailed descriptions are necessary. Below are a few specific questions:

Fig. 3: In panel (b), what are the different lines and points representing?

Fig. 4 and Fig. 5: Why are the red and blue dots there? H monomers or P monomers?

Fig. 6: In panel (b), what do the symbols (Au, Af, etc.) mean??

2. The relationships between the different sections are not clear. Table I may be moved to the main text and further discussion of the correspondence between DEM cycle and HP foldamer model may be provided.

3. Page 4, Line 141. “Folded proteins are nanoscale solids, different for different sequences. In contrast, because RNA molecules are stiff and hydrogen bonded, they tend to be stringier - less folded, less compact and more floppy - and with poorly defined sequence-to-structure relationships.”

Structures are generally more conserved than the sequences and different RNA sequences can fold to the same or similar structures. Indeed, RNAs are usually more flexible. It would be useful for the authors to clarify the meaning of “poorly defined sequence-to-structure relationships” for RNAs.

4. Page 4, Line 144. “Accurate self-copying is not the property we seek here.”

Why is an accurate replication of informational polymers not a key property for the autocatalytic system? More discussions would be helpful here.

5. Page 4, the last paragraph describes the “golf course” landscape and an idealized “funnel” landscape for fast protein folding. What about more complex funnel landscapes, such as a global funnel-like landscape that involves kinetic traps and bumps? Can complex funnel landscapes also result in disorder-to-order transformation?

6. Page 5, Line 173. “However, it has been found in computer modeling that some heteropolymers behave differently.”

Citations of references are needed here.

7. Page 6, Line 186. “These landing pads on foldcats could catalyze the covalent elongation of other “client” HP sequences.”

In the HP foldamer mechanism, how long can a sequence be elongated? How does the covalent elongation occur when the H monomer is close to the landing pads on the foldcats?

8. The key idea behind the HP foldamer mechanism comes from the hydrophobic effects, where the hydrophobic “landing pads” can attract peptides and catalyze polymerization. The “landing pads” idea is quite interesting and may be generalized in the discussion. For example, nucleotides in a loop region of a folded RNA may also serve as the “landing anchors” for other nucleotides/short chains through base pairing.